# Weakly supervised lung cancer detection on label-free intraoperative microscopy with higher harmonic generation

**Siem de Jong**[1]      SIEM.DEJONG@RADBOUDUMC.NL

**Marie Louise Groot**[2]      M.L.GROOT@VU.NL

**Roel L. J. Verhoeven**[3]      ROEL.LJ.VERHOEVEN@RADBOUDUMC.NL

**Erik H. F. M. van der Heijden**[3]      ERIK.VANDERHEIJDEN@RADBOUDUMC.NL

**Francesco Ciompi**[1]      FRANCESCO.CIOMPI@RADBOUDUMC.NL

[1] *Radboud university medical center, Department of Pathology, Nijmegen, The Netherlands*

[2] *Vrije Universiteit Amsterdam, Faculty of Science, Department of Physics, LaserLab, Amsterdam, The Netherlands*

[3] *Radboud university medical center, Department of Pulmonary Diseases, Nijmegen, The Netherlands*

## Abstract

Higher harmonic generation microscopy (HHGM) enables label-free on-site imaging of fresh tissue, potentially allowing a new means of pathology assessment for disease diagnosis. We investigate the potential of using self-supervised learning (SSL) in combination with weakly-supervised, attention-based, clustering constrained multiple instance learning (CLAM) to detect lung cancer in HHGM images. First, we tailor encoders to HHGM-specific data domain via both SimCLR and DINO SSL. Second, we train a CLAM classifier with and without an SSL feature extractor on 100 HHGM images acquired during bronchoscopy procedures. We show that SSL pre-training with random initialization and CLAM are beneficial to intraoperatively detect lung cancer in HHGM images.

**Keywords:** higher harmonic generation microscopy, self-supervised pre-training, weakly-supervised learning, classification, attention, explainability, intraoperative, lung cancer

## 1. Introduction

Lung cancer is the second most deadly cancer in the Netherlands with more than 10.000 deaths in 2022 (Nederlandse Kankerregistratie, IKNL, 2024). Histopathology is at the core of diagnosis and treatment planning, but implies time-consuming processes that are becoming increasingly more difficult with the expected shortage on pathologists (Bychkov and Fukuoka, 2022). Additionally, intraoperatively during the bronchoscopy procedure, tumor detection has to be done quickly to optimally guide the next step and avoid longer procedures and increased patient discomfort. To decrease the time needed for tissue sample assessment, higher harmonic generation microscopy (HHGM) might be used (Van Huizen et al., 2020). The label-free HHGM images consist of second and third harmonic generation (SHG, THG) and two-photon excitation fluorescence (2PEF). SHG shows non-centrosymmetric molecules, such as collagen, THG is generated by local differences of cell and structure interfaces, and 2PEF is produced by endogenous fluorophores such as elastin.

To aid on-site tumor detection, we developed a deep learning model for tumor detection and investigated the potential of using self-supervised learning (SSL) in combination with a weakly-supervised, attention-based, clustering constrained multiple instance learning (CLAM) (Lu et al., 2021). The attention-based classifier can learn to focus on the most important parts of the image and use instance clustering to constrain and refine the search space, which is useful since only image level annotations are available. Out of the box,

CLAM uses an ImageNet pre-trained backbone, but the domain of natural images is very different from HHGM. Inspired by Saillard et al. (2021) and Schirris et al. (2022), we show that employing SSL pre-training to obtain domain-specific features in combination with a CLAM classifier improves lung cancer detection on HHGM.

## 2. Methods

**Data.** We use HHGM images acquired in AmsterdamUMC from October 2020 until November 2021 (Van Huizen et al., 2024). Bronchoscopy or thoracoscopy patients with suspected lung cancer or pleural malignancy were eligible for the study. The dataset consists of 100 individually assessed HHGM images at 0.5 mpp of 47 cases with multiple biopsies and scans at multiple depths. SHG, THG, and 2PEF signals were mapped to the red, green, and blue channels of RGB data, respectively (Figure 1C). Image-level ground truth as presence or absence of tumor was provided by two pathologists. The tissue was segmented using EntropyMasker (Song et al., 2022) and tiled into 256x256 pixel images.

**Self-supervised pre-training.** We used the extracted patches to pre-train four ResNet-18 models via self-supervision. In two models, we used SimCLR (Chen et al., 2020) with normalized temperature cross-entropy loss and in two other models, we used DINO (Caron et al., 2021). For each pre-training approach, we initialized one model with ImageNet pre-trained weights and the other with He initialization (He et al., 2015). We augmented tiles with random resized crop, random horizontal flip, color jitter (brightness, contrast, saturation, hue), and random solarization for DINO-based SSL. The SimCLR projection head has three fully connected layers of dimensions 1024, 1024, and 64. The DINO projection head has three fully connected layers of 512, 64, and 256. Stochastic gradient descend (SGD) was used with learning rate 0.1 and momentum 0.001. The pre-trained models were used to extract tile features of size 512 for the classifier.

**Classification.** CLAM was trained using the slide-level label to classify tile features as tumor or non-tumor. The attention network has two gated attention layers of size 128 and 64. CLAM was trained with SGD with a minimum of 100 epochs, early stopping, dropout with $p = 0.25$, and a learning rate of $1 \times 10^{-6}$. We leverage CLAM attention maps to gain insight into areas important for the prediction.

**Evaluation.** The existing dataset is split in 45 training (86 images) and 2 test cases (14 images). On the training data, we perform 10-fold Monte Carlo cross-validation stratified by outcome and grouped by patient. Receiver operating characteristic (ROC) curves were interpolated, yielding a mean ROC curve. The area under the ROC curve (AUC) was calculated for the mean ROC curves. After the cross-validation procedure, for evaluation on the test set, we trained a new model using the learning setup with the highest cross-validation AUC, using the complete training set for 100 epochs. Statistical bootstrapping with 9999 predictions was used to obtain a mean test AUC and 95 % tolerance interval (TI).

## 3. Results and discussion

The validation ROC curves are shown in Figure 1A. The SimCLR- and DINO pre-trained models have a higher AUC than a model only pre-trained on ImageNet. The ImageNet dataset shows features dissimilar to HHGM signals. The higher AUC for models with self-supervised pre-training suggest that pre-training on HHGM images is beneficial for CLAM.

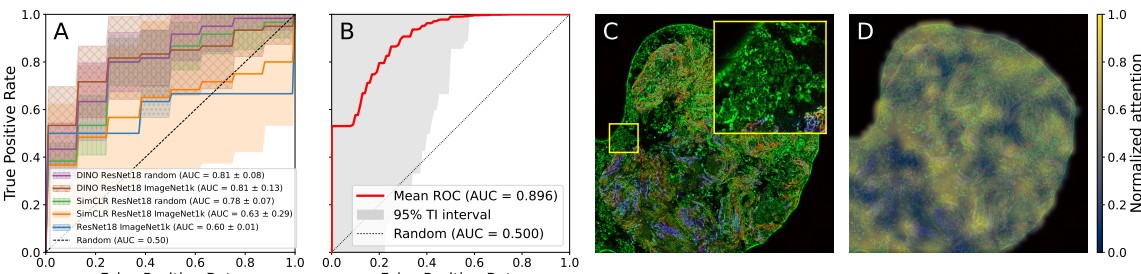

Figure 1: A) Validation mean ROC curves with ±1 standard deviation per model across folds. B) Hold out test ROC curve with 95 % TI of the proposed random initialized DINO pre-trained ResNet18 model. C) An original HHGM test image. Inset shows a zoomed-in cell-rich region. D) CLAM attention map acquired by the proposed model.

The He initialized model trained with SimCLR has a +23.8 % AUC increase compared to fine-tuning an ImageNet pre-trained feature extractor. When using DINO, the He and ImageNet initialized perform comparably. However, the standard deviation of the DINO model is smaller. These results suggest that training a randomly initialized model with self-supervision increases CLAM performance, but they also show that ImageNet initialized models can be adapted to the target domain with self-supervision.

DINO pre-trained models seem to outperform SimCLR pre-trained models on AUC, which might be a result of the DINO momentum encoder.

The DINO-pre-trained He initialized model has the highest AUC of $0.81 \pm 0.13$. Therefore, we retrain this model on the complete training set and evaluate the model on the held out test set. Statistical bootstrapping resulted in the test ROC curve shown in Figure 1B and a mean test AUC of 0.896 (95 % TI 0.712 to 1.000). An example of a CLAM heatmap next to an HHGM image is shown in Figure 1C-D. The attention maps show that the model focuses on the tumor regions as expected, but also on non-tumor areas, limiting the interpretation. CLAM is a step towards interpretable models, but separating attention maps per class like in Javed et al. (2022) might increase explainability.

A limitation of this study is the low amount of test data which also only includes invasive adenocarcinoma and chronic inflammation. More data is being acquired to further train the model and validate it on more cases with more variety. However, the results are promising and suggest that it might be possible in the future to automatically detect lung cancer intraoperatively using HHGM and artificial intelligence.

## Acknowledgements

We are extremely grateful to Van Huizen et al. (2024) for sharing their dataset. Many thanks to Laura M. G. van Huizen, Jouke T. Annema, and Johannes M. A. Daniels for their work on validating HHGM in the clinic. We are also grateful to Jan H. von der Thüsen and Teodora Radonic for providing the ground truth. This study was funded by the European Union under the IMAGIO grant number 101112053. Views and opinions expressed are however those of the authors only and do not necessarily reflect those of the European Union or Innovative Health Initiative Joint Undertaking. Neither the European Union nor the granting authority can be held responsible for them.

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
