# OpenReview forum: "Weakly supervised lung cancer detection on label-free intraoperative microscopy with higher harmonic generation"
_MIDL.io/2024/Short_Papers — MIDL 2024 Short Papers_

### Official Review · Reviewer_31wq · 2024-04-18

**Confidence:** 4
**Final Rating:** 4

**Review:**

The paper addresses detection of lung cancer in higher harmonic generation microscopy scans, the method is based on self-supervised learning and weakly supervised learning.

Strengths:
-	Novel application (or at least one that you don’t see very often, my expertise is in methodology)
-	Clear descriptions of data and method
-	Interesting to investigate different types of pretraining and not just go with a default
-	AUC variability reported in experiments

Weaknesses:
-	Quite a lot of results packed into a small space, it is difficult to appreciate the differences between the methods.
-	Test set is small, but this is acknowledged.

I think this is an interesting and relevant preliminary study so would recommend it being accepted for the conference.

---

### Decision · Program_Chairs · 2024-04-26

Accept